# Modeling the Effects of Climate Change and Land Use/Land Cover Change on Sediment Yield in a Large Reservoir Basin in the East Asian Monsoonal Region

Huiyun Li [1], Chuanguan Yu [2], Boqiang Qin [1,*], Yuan Li [3], Junliang Jin [4], Liancong Luo [5,*] [ID], Zhixu Wu [2], Kun Shi [1] and Guangwei Zhu [1]

[1] State Key Laboratory of Lake Science and Environment, Nanjing Institute of Geography and Limnology, Chinese Academy of Sciences, Nanjing 210008, China; hyli@niglas.ac.cn (H.L.); kshi@niglas.ac.cn (K.S.); gwzhu@niglas.ac.cn (G.Z.)
[2] Hangzhou Bureau of Ecology and Environment Chun'an Branch, Hangzhou 311700, China; ycggates@126.com (C.Y.); caepb@126.com (Z.W.)
[3] School of Tourism and Urban & Rural Planning, Zhejiang Gongshang University, Hangzhou 310018, China; liyuan@zjgsu.edu.cn
[4] Yangtze Institute for Conservation and Development, Nanjing Hydraulic Research Institute, Nanjing 210098, China; jljin@nhri.cn
[5] Institute for Ecological Research and Pollution Control of Plateau Lakes, School of Ecology and Environ-Mental Science, Yunnan University, Kunming 650504, China
* Correspondence: qinbq@niglas.ac.cn (B.Q.); billluo@ynu.edu.cn (L.L.)

**Abstract:** This research addresses the separate and combined impacts of changes in climate and land use/land cover on the hydrological processes and sediment yield in the Xin'anjiang Reservoir Basin (XRB) in the southeast of China by using the soil and water assessment tool (SWAT) hydrological model in combination with the downscaled general circulation model (GCM) projection outputs. The SWAT model was run under a variety of prescribed scenarios including three climate changes, two land use changes, and three combined changes for the future period (2068–2100). The uncertainty and attribution of the sediment yield variations to the climate and land use/land cover changes at the monthly and annual scale were analyzed. The responses of the sediment yield to changes in climate and land use/land cover were considered. The results showed that all scenarios of climate changes, land use/land cover alterations, and combined changes projected an increase in sediment yield in the basin. Under three representative concentration pathways (RCP), climate change significantly increased the annual sediment yield (by 41.03–54.88%), and deforestation may also increase the annual sediment yield (by 1.1–1.2%) in the future. The comprehensive influence of changes in climate and land use/land cover on sediment yield was 97.33–98.05% (attributed to climate change) and 1.95–2.67% (attributed to land use/land cover change) at the annual scale, respectively. This means that during the 2068–2100 period, climate change will exert a much larger influence on the sediment yield than land use/land cover alteration in XRB if the future land use/land cover remains unchanged after 2015. Moreover, climate change impacts alone on the spatial distribution of sediment yield alterations are projected consistently with those of changes in the precipitation and water yield. At the intra-annual scale, the mean monthly transported sediment exhibits a significant increase in March–May, but a slight decrease in June–August in the future. Therefore, the adaptation to climate change and land use/land cover change should be considered when planning and managing water environmental resources of the reservoirs and catchments.

**Keywords:** climate change; land use/land cover change; sediment response; multiple scenarios; modeling

## 1. Introduction

Catchment sediment yield is mainly controlled by soil properties, topography, climate condition, and land use/land cover types [1–4]. In contrast, the soil properties and topogra-

phy are relatively stable, while climate and land use/land cover are variable over a specific time period [5,6]. Climate change, mainly in the form of temperature and precipitation, has a direct impact on runoff and an indirect impact on sediment by changing the process of the water cycle in the basin, and further influences the phytoplankton community [7–9]. Land use/land cover changes caused by anthropogenic activities may re-distribute the rainfall-runoff by changing the processes of infiltration, evapotranspiration, and groundwater recharge, which has a profound impact on the water and sediment production mechanism [10,11].

The impacts of climate change on streamflow and sediment yield have been investigated in a number of studies [12–18]. A previous study indicated that the runoff increased by 1.3% and the sediment yield increased by 2% for every 1% increase in rainfall in eight large Chinese catchments [12]. Similarly, a preliminary study of a watershed in Spain showed that higher precipitation is usually associated with more runoff and soil loss [13]. This is not only because precipitation increases soil moisture, but also because it saturates soil moisture or produces soil crusts [14]. In contrast, Zhao et al. showed that the reduction in precipitation was one of the main factors, leading to the sharp reduction in the discharge and sediment yield in the middle reaches of the Yellow River [15]. However, the impacts of precipitation change on soil erosion are complicated and are not always negative. Increasing rainfall may increase the plant biomass and vegetation canopy, thus reducing the runoff and erosion [16]. In addition to precipitation, temperature is also one of the important meteorological factors affecting the sediment of the basin [17,18]. For example, Syvitski divided the watersheds into climatic zones according to different temperatures, and found that the average temperature of the watershed has an important impact on sediment transport [18].

On the other hand, the joint effects of climate variability and vegetation change on hydrological process have been a key research point. Such synergistic influences on hydrological processes and sediment yields are complex [19]. Some studies have found that sediment alteration was dominantly influenced by land use/land cover changes, while some showed that climate variability was a more important impact factor [20]. It is essential to accurately distinguish and quantify the effects of climate variability/climate change on streamflow and sediment for catchment and reservoir management in the future under different conditions [21–24]. Compared to the influence on streamflow, few works have concerned the sediment spatial and temporal changes in response to combining the variations in the land use/land cover with climate change for an uncertain future. Therefore, a thorough study on the impacts of multiple climatic conditions and land use/land cover scenarios on sediment is needed [25].

An IPCC Special Report stated that a global warming of 1.5 °C above pre-industrial levels has significantly affected the hydrological process including the quality and quantity of water resources in many regions [26,27]. Until now, numerous studies on assessing the response of hydrological circles to climate-driven force have widely applied the general circulation model (GCM) projections of the coupled model inter-comparison project phase 5 (CMIP5) [4,28]. A tentative conclusion is that RCP2.6, RCP4.5, RCP6.0, and RCP 8.5 are responsible for a 16.3%, 14.3%, 36.7%, and 71.4% increase in future streamflow, and a 16.5%, 32.4%, 81.8%, and 170% increase in future sediment yield, respectively, in northeastern China [4]. An increase in monthly streamflow (maximum increases by 52–170% under different RCP scenarios) was reported, along with a monthly average decrease in sediment concentrations of 10% projected in southwest Iran in the future [28]. Although GCM outputs have been extensively employed to study the impacts of climate change on the hydrological process in many locations, it is problematic to use GCM outputs directly in hydrological models at regional and local scales because of the low resolution of GCM projections [29]. Therefore, downscaling methods are often applied to obtain regional scale analysis of meteorological variables from coarse-scale GCM outcomes to allow the conclusions on streamflow and sediment regime changes to be more reliable [30].

The Xin'anjiang Reservoir, which is the largest reservoir in the Yangtze River Delta in China, plays quite an important role in the local water supply, fishery, water transportation, and crop irrigation [31]. The Xin'anjiang Reservoir is famous for its excellent water quality; however, the pressure of water environment protection in the reservoir is increasing year by year [32]. The Jiekou section, located in the estuary area of the Xin'anjiang Reservoir, in particular, is facing the problem of a decrease in water transparency and the risk of algal blooms [33]. This might be related to the climate variability and land use/land cover change in the basin. Previous studies have noted that the annual streamflow through the Jiekou section, accounting for over 60% of the total inflows of the Xin'anjiang Reservoir, showed an obvious increasing trend in the last few decades caused by rainstorms [34,35]. However, few studies have attempted to identify how climate variability and land use/land cover change affect sediment yield. In this study, we focused on identifying and quantifying the effects of climate change and land use/land cover change on the sediment yield using a hydrological modeling approach. With the help of our research results, a deeper understanding of sediment response to climate-driven forcing and land use/land cover changes in XRB would be beneficial for water quality protection and bloom prevention of the reservoir in the East Asian monsoonal region.

## 2. Data and Methods

### 2.1. Study Area

The Xin'anjiang River drains into the Xin'anjiang Reservoir, Chun'an, Zhejiang Province, southeast China, situated within a watershed that spans an area of roughly 10,442 km$^2$ (Figure 1) [36]. The reservoir has a surface area of 573 km$^2$ and a water storage capacity of 178.4 × 10$^8$ m$^3$ when the normal water storage level is 10$^8$ m asl [37]. The longest path of the river is over 370 km, and two river gauging stations are located at Tunxi and Yuliang, respectively. The basin is dominated by a typical subtropical humid monsoon climate and enters the East Asian rainy season, also known as the plum rain, in June and July every year [38]. For the last 50 years, the mean annual precipitation has been about 1621 mm, the mean annual runoff is about 1018 mm, and the mean annual air temperature has ranged from 16.7 °C to 18.9 °C. Approximately 42% of the annual precipitation is contributed by monsoons (June–September), and the maximum humidity is recorded as 100% in June and July.

Jiekou is the main entrance for the streamflow and sediment of the Xin'anjiang River to Xin'anjiang Reservoir by controlling around 60% of the area of the whole basin [39]. The elevation of the basin varies from −1 m to 1764 m from the mean sea level. The terrain is complex and diverse with mainly a geomorphic type of mountains. The zonal soil types of the basin are mainly red soil, yellow soil, and yellow brown soil, which are distributed vertically according to the altitude. The area is covered with dense forests, which is the most widely distributed land-use type. The cultivated land is concentrated at the periphery of urban land [40].

### 2.2. Data Description

#### 2.2.1. Hydrometeorological Data

Daily meteorological data recorded including air temperature (°C), precipitation (mm), relative humidity (%), solar radiation (MJ/m$^2$/day), and wind speed (m/s) from 1973 to 2018 at two meteorological stations (Figure 1) were downloaded from the website of the National Meteorological Information Center (China Meteorological Administration, CMA) (http://data.cma.cn/en (accessed on 1 January 2019)) [41]. The observed daily streamflow data for the period of 2001–2014 at two hydrological stations (Figure 1) were collected from the Hydrological Data Yearbook published by the Ministry of Water Resources of the People's Republic of China (MWR) [42]. The mean sediment transport rate investigated from 2006 to 2014 was obtained from the same data source. The time series above was checked for outliers and errors in order to be used in hydrological modeling.

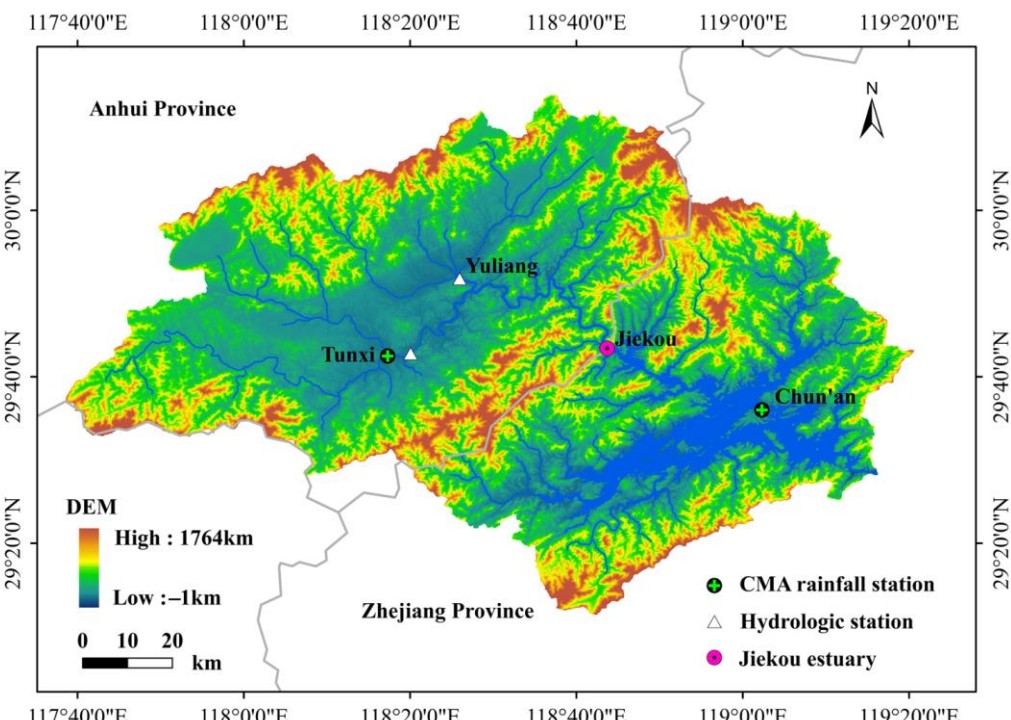

**Figure 1.** The location of the Xin'anjiang Reservoir Basin, China.

### 2.2.2. Geospatial Data

The basic geospatial datasets required to construct the model include a digital elevation model (DEM), a soil classification map, and land use information. The DEM map with a 90 m spatial resolution used for watershed delineation and sub-basin discretization was downloaded by the Geospatial Data Cloud of China. The 1 km resolution soil map was originally derived from the Harmonized World Soil Database (HWSD), which is produced by the Food and Agriculture Organization of the United Nations [43]. The soil data over China were derived from the results of the Second National Land Survey organized by China's State Council from 2007 to 2009. This was produced by the Institute of Soil Science, Chinese Academy of Sciences.

To estimate the effect of land use/land cover change, two land use/land cover maps with a spatial resolution of 30 m for XRB were interpreted from Landsat imagery in 1987 and 2015, respectively. The cloud was masked based on the pixel_qa band of the Landsat surface reflectance data (https://developpers.google.com/earth-engine/datasets/catalog/LANDSAT_LC08_C01_T1_SR (accessed on 1 May 2020)) after the images were obtained. A median imagery was output by calculating the median value at each pixel of all images in one collection (https://developpers.google.com/earth-engine/reducers_image_collection (accessed on 1 May 2020)). For each median imagery, the land use/land cover information was extracted with a support vector machine classification algorithm in the ENVI (version 5.3). The land use/land cover here is classified into five classes for the SWAT model, namely forest, water body, cultivated land, urban land, and bare land [44].

### 2.2.3. RCP Data

Representative concentration pathways (RCPs) including a stringent mitigation scenario (RCP2.6), an intermediate scenario (RCP4.5), and one scenario with very high GHG emissions (RCP8.5) [45] were used to estimate the impacts of climate change. Eighteen datasets obtained from a coupled model inter-comparison project phase 5 (CMIP5) GCM for XRB were downloaded from the website of the World Climate Research Program (https://esgf-node.llnl.gov/search/cmip5/ (accessed on 1 February 2020)). The Taylor diagram method was adopted to assess the performance of datasets from CMIP5 GCMs in simulating the historical meteorological elements [46]. Four assessment criteria—the

correlation coefficient ($r$), root mean square error ($RMSE$), standard deviation of observed values ($\sigma_O$), and standard deviation of simulated values ($\sigma_S$)—were used to identify the most applicable dataset. More detailed information about the Taylor diagram method can be found in Taylor [46].

The selected CMIP5 datasets comprise three meteorological elements (daily air temperature and precipitation) for a historical period (1901–2005) and a projection period (2006–2100, RCP2.6, RCP4.5, and RCP8.5 scenarios). The original resolution data were downscaled into $0.5° \times 0.5°$ by the China Meteorological Data Service Center (CMDC) using a statistical downscaling method.

### 2.3. Methodology

An integrated framework was designed to evaluate the effect of climate change and land use/land cover change on the streamflow and sediment yield using XRB as a case study. To set up the structure of this approach, we (1) assessed the accuracy and availability of the downscaled GCM data, and the interpreted land use/land cover map from remote sensing imagery; (2) designed individual and combined climate and land use/land cover change scenarios; (3) modeled streamflow and sediment yield response under uncertainty; and (4) evaluated the streamflow and sediment variation under climate change and land use/land cover change. The simulation baseline is in the period of 1973–2005 and the future is in the period of 2068–2100.

#### 2.3.1. Climate Change and Land Use/Land Cover Change Scenarios

Three RCP scenarios were selected in this study to assess how different emissions impact streamflow and sediment yield, namely, RCP2.6, RCP4.5, and RCP8.5. These three scenarios represent the total radiative force in 2100 relative to pre-industrial values, which are +2.6, +4.5 and +8.5 W/m$^2$, respectively. The calibrated SWAT model was used to simulate the following eight scenarios: $SN_B$, $SN_{2.6}$, $SN_{4.5}$, $SN_{8.5}$, $SN_B^{LC}$, $SN_{2.6}^{LC}$, $SN_{4.5}^{LC}$, and $SN_{8.5}^{LC}$, respectively, as listed in Table 1. The $SN_B^{LC}$, $SN_{2.6}^{LC}$, $SN_{4.5}^{LC}$, and $SN_{8.5}^{LC}$ were essentially the $SN_B$, $SN_{2.6}$, $SN_{4.5}$, and $SN_{8.5}$ scenarios with the addition of the land use/land cover change. We present the land utilization condition in 1987 as the baseline of the land use/cover, based on assuming that there were no significant changes in the land use/land cover during the baseline period (1973–2005). Similarly, we used the land use/land cover map in 2015 as the representative land use/land cover in the future period (2068–2100). We assumed that there would be no significant changes in the land use/land cover between the future period (2068–2100) and that in 2015. More details on our scenarios can be found in Table 1.

**Table 1.** The scenario analysis for characterizing the effects of climate change and land use/land cover change on the streamflow and sediment.

| Scenario | Simulation Time | Land Use/Cover | Climate | Description |
|---|---|---|---|---|
| $SN_B$ | 1973–2005 | LULC1987 | History | Baseline |
| $SN_{2.6}$ | 2068–2100 | LULC1987 | RCP2.6 | With a stringent mitigation scenario and no land use/land cover change |
| $SN_{4.5}$ | 2068–2100 | LULC1987 | RCP4.5 | With an intermediate scenario and no land use/land cover change |
| $SN_{8.5}$ | 2068–2100 | LULC1987 | RCP8.5 | With a very high greenhouse gas emission scenario and no land use/land cover change |
| $SN_B^{LC}$ | 1973–2005 | LULC2015 | History | With a land use/land cover change and no climate change |
| $SN_{2.6}^{LC}$ | 2068–2100 | LULC2015 | RCP2.6 | With land use/land cover change and a stringent mitigation scenario |
| $SN_{4.5}^{LC}$ | 2068–2100 | LULC2015 | RCP4.5 | With land use/land cover change and intermediate scenario |
| $SN_{8.5}^{LC}$ | 2068–2100 | LULC2015 | RCP8.5 | With land use/land cover change and a very high greenhouse gas emission scenario |

### 2.3.2. SWAT Hydrological Model

The SWAT (Soil & Water Assessment Tool) model developed by the USDA (the United States Department of Agriculture) is a semi-distributed, process-based, continuous, daily time-step hydrological model. It has been widely applied to represent the main hydrological processes within small and large basins [11,25,47]. In this study, ArcSWAT (an ArcGIS-ArcView extension and interface for SWAT) running on the ArcGIS (version 10.2) platform as an interface was used to assess the streamflow and sediment yield. Several sub-basins and multiple HRUs (Hydrologic Respond Units) are divided according to the land use types, soil classes, and slopes. Erosion caused by rainfall and runoff is calculated with the Modified Universal Soil Loss Equation (MULSE) in the SWAT model [48]. Parameter sensitivity analysis, calibration, and validation are carried out by SWAT-CUP (SWAT Calibration and Uncertainty Programs), which is an automatic sensitivity analysis tool in the SWAT model [49–51]. Sensitivity analysis is the procedure used to identify the most influential parameters for calibration using SUFI-2 (the global sensitivity analysis of the sequential uncertainty fitting) algorithm. In order to evaluate the performance of the SWAT model in streamflow and sediment yields simulations, the Nash–Sutcliffe coefficient of efficiency ($NSE$) and coefficient of determination ($r^2$) between the observed and estimated values were calculated by [52]:

$$NSE = 1 - \frac{\sum_{i=1}^{n}(O_i - S_i)^2}{\sum_{i=1}^{n}(O_i - \overline{O})^2} \tag{1}$$

$$r^2 = \frac{\left[\sum_{i=1}^{n}(O_i - \overline{O})(S_i - \overline{S})\right]^2}{\sum_{i=1}^{n}(O_i - \overline{O})^2 \sum_{i=1}^{n}(S_i - \overline{S})^2} \tag{2}$$

where $O_i$ and $S_i$ are the observed and simulated hydrological parameters and $\overline{O}$ and $\overline{S}$ are the mean of observed and simulated values, respectively. The criterion considers the model performance to be: very good if $0.75 \leq NSE < 1.00$ and $r^2 = 1.00$; good if $0.65 < NSE \leq 0.75$ and $0.80 \leq r^2 < 1.00$; satisfactory if $0.40 < NSE \leq 0.65$ and $0.50 \leq r^2 < 0.80$; unsatisfactory if $NSE \leq 0.40$ and $r^2 < 0.50$ [53–55].

### 2.3.3. Sediment Response to Changes of Climate and Land Use/Land Cover

For a given catchment, the total change in the mean annual sediment between independent periods with different climatic RCP scenarios and land use/land cover characteristics can be estimated as:

$$\Delta D_{RCPj}^{LC} = \Delta D_{RCPj} + \Delta D^{LCj}, \; j = 2.6, \; 4.5 \text{ and } 8.5 \tag{3}$$

where $\Delta D_{RCPj}^{LC}$ indicates the total change in the mean annual sediment between the future and baseline and $\Delta D_{RCPj}$ is the change in the mean annual sediment because of the climate change (different RCP scenarios, $j = 2.6$, 4.5 and 8.5, respectively) between the two periods. We assumed that there were almost no other regulations or diversions except for land use/land cover change in the catchment. $\Delta D^{LCj}$ indicates the change in the mean annual sediment as a result of change in the land use/land cover change between the two periods.

To separate the sediment yield impacts caused by climate variability and land use/land cover change, an effective method used to quantify $\Delta D_{RCPj}$ and $\Delta D^{LC}$ can be seen in the following expressions [56]:

$$\Delta D_{RCPj} = \frac{(D_{RCPj} - D_B) + \left(D_{RCPj}^{LC} - D^{LC}\right)}{N}, \; N = 2, \; j = 2.6, \; 4.5 \text{ and } 8.5 \tag{4}$$

$$\Delta D^{LCj} = \frac{(D^{LC} - D_B) + \left(D_{RCPj}^{LC} - D_{RCPj}\right)}{N}, \; N = 2, \; j = 2.6, \; 4.5 \text{ and } 8.5 \tag{5}$$

where $D_{RCPj}$ ($j$ = 2.6, 4.5, and 8.5) are the mean annual sediment under the RCP2.6, RCP4.5, and RCP8.5 scenarios, respectively, with the historical land use/land cover condition. $D_B$ is the mean annual sediment of the baseline. $D_{RCPj}^{LC}$ is the mean annual sediment under different RCP scenarios after catchment land use/land cover change. $D^{LC}$ indicates the mean annual sediment caused by land use/land cover change without climate variability. The climate condition when simulating $D^{LC}$ was as the same as simulating $D_B$.

As a result of the above, the difference in sediment between RCP scenarios ($D_{RCPj}$) and the baseline ($D_B$) can be considered as the impacts of RCP scenarios on sediment change (1987 land use/land cover condition). Similarly, the difference in sediment between $D_{RCPj}^{LC}$ and $D^{LC}$ can be considered as the impacts of RCP scenarios on sediment change (2015 land use/land cover condition). On the other hand, the effects of land use/land cover change on sediment can be determined by applying the difference between $D^{LC}$ and $D_B$ or between $D_{RCPj}^{LC}$ and $D_{RCPj}$. The difference between sediment in different RCP scenarios after land use/land cover change (2015 land use/land cover condition) and the baseline represents the combined effects of climate variability and land use/land cover change. The combined effects can also be described as:

$$\Delta D_{RCPj}^{LC} = D_{RCPj}^{LC} - D_B, \ j = 2.6, \ 4.5 \text{ and } 8.5 \tag{6}$$

Therefore, the percentage contributions of different RCP scenarios ($\alpha_{RCPj}$) and land use/land cover change ($\alpha^{LC}$) to the variations in sediment can be expressed by:

$$\alpha_{RCPj} = \frac{\Delta D_{RCPj}}{\Delta D_{RCPj}^{LC}} \times 100\%, \ j = 2.6, \ 4.5 \text{ and } 8.5 \tag{7}$$

$$\alpha^{LCj} = \frac{\Delta D^{LCj}}{\Delta D_{RCPj}^{LC}} \times 100\%, \ j = 2.6, \ 4.5 \text{ and } 8.5 \tag{8}$$

## 3. Results and Discussion

### 3.1. Climate Change Analysis under Varying Scenarios

Monthly meteorological data from CMA were used to assess the performance of GCM outputs in climate in XRB. The arithmetic average value of the records of Tunxi Station and Chun'an Station represented the average value of the basin. Eighteen meteorological datasets including three elements (maximum temperature, minimum temperature, and precipitation) from downscaled CMIP5 GCMs were used to plot a Taylor diagram against the CMA data (see Figure 2). For the monthly minimum temperature, $r$ values between the CMA data and eighteen GCM outputs were 0.94–0.97, and all $RMSE$ values were less than 2.5 °C. Meanwhile, the $\sigma_S$ of all GCM monthly minimum temperatures and $\sigma_O$ were very close. Hence, all eighteen GCMs were suitable to simulate the historical data of the monthly minimum temperature (1973–2005). Eleven datasets outperformed the other GCM outputs for the monthly maximum temperature with higher $r$ and lower $RMSE$ (Figure 2). It can be seen from the Taylor diagram that the simulation results of all eighteen GCMs on monthly precipitation were not as good as those on monthly temperature. The highest $r$ for precipitation was around 0.38, and the lowest $RMSE$ was around 8.0 mm. Taken together, a certain dataset, namely, CSIRO-Mk3-6-0, was selected to evaluate the effect of climate change between the future and historical periods due to its best performance in climate simulation in the basin.

Figure 3a,c,e shows the time series of the downscaled CSIRO-Mk3-6-0 annual maximum and minimum temperature averaged over XRB in the baseline period (1973–2005) and 2006–2100. The mean annual maximum and minimum temperatures of the basin at the baseline were 20.79 °C and 11.43 °C, respectively, while those in the last 33 years of the 21st century (the simulation period, 2068–2100) will be increased dramatically by 1.91–5.11 °C and 1.75–4.46 °C relative to that at the baseline. The mean monthly maximum temperature

and minimum temperature from 2068 to 2100 under RCP2.6, RCP4.5, and RCP8.5 will be increased by 1.36–7.14 °C and 0.72–7.18 °C relative to that at the baseline. XRB has four distinctive seasons, with the highest increases in seasonal maximum and the minimum temperature of 2.46–5.65 °C and 2.59–5.97 °C in fall (September–November) under different RCPs, respectively.

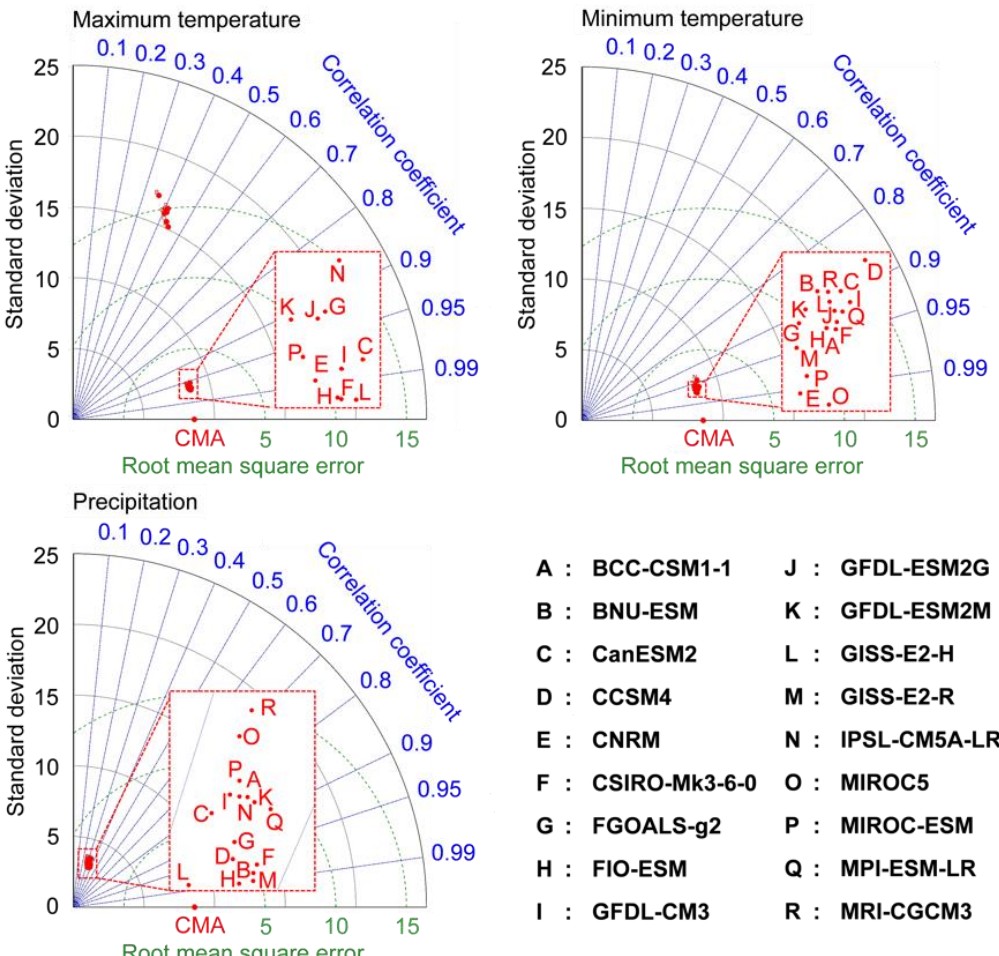

**Figure 2.** The Taylor diagram of the monthly maximum temperature, minimum temperature, and precipitation simulated by the 18 GCM models.

The basin has a subtropical monsoon climate, and the precipitation is significantly affected by monsoon circulation. Figure 3e shows that the mean annual precipitation in 1973–2005 was 1662.48 mm, while that in 2068–2100 increased significantly by 97.69–285.72 mm relative to that in 1973–2005 under RCP2.6, RCP4.5, and RCP8.5. The temporal distribution of precipitation in XRB is nonuniform. The precipitation from spring (March–May) and summer (June–August) accounted for 36.78% and 35.36% of the total precipitation in a year, respectively. The precipitation was low in the fall and winter seasons from September to February of the next year, which accounted for 27.86% of the total precipitation in a year. In 2068–2100, no significant changes in precipitation were observed in the spring, fall, and winter seasons. There was abundant precipitation in summer (June–August), with an increase of 4.79–9.35% under three RCP scenarios relative to the same period in 1973–2005. The most obvious increase in precipitation occurred in June and accounted for 19.45–35.99% of the same month in 1973–2005 under different RCPs.

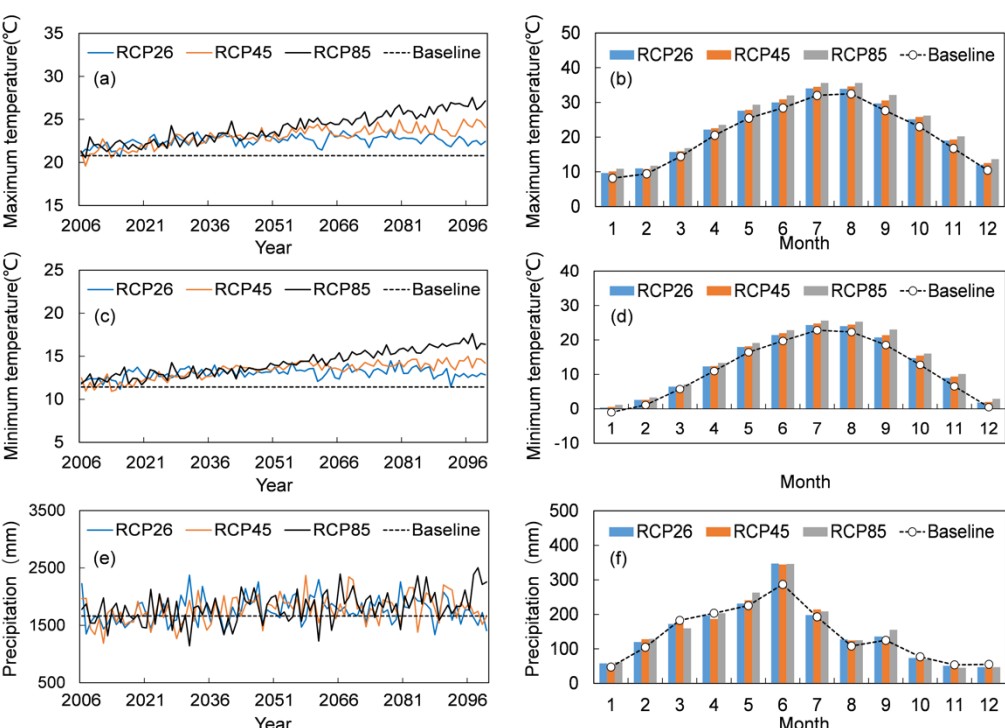

**Figure 3.** The time series of annual maximum temperature (**a**), annual minimum temperature (**c**), annual precipitation (**e**), monthly maximum temperature (**b**), monthly minimum temperature (**d**), and monthly precipitation (**f**) averaged over XRB projected by downscaled CMIP5 GCM (CSIRO-Mk3-6-0) in 2006–2100 under RCP2.6, RCP4.5 and RCP8.5, respectively. The solid lines and histograms indicate CSIRO-Mk3-6-0 outputs of different RCPs, the dotted line is the mean values of annual meteorological elements of the basin in baseline (1973–2005). The dashed line with circles indicates the mean values of monthly meteorological elements.

### 3.2. Land Use/Land Cover Change Analysis under Varying Scenarios

The land use/land cover classification map of 1987 and 2015 were interpreted from Landsat imagery (see Figure 4). Take the patterns of land use in 1987 as the representative underlying surface type in the baseline period, while the patterns of land use in 2015 represent the underlying surface type condition in the future. It is assumed that there will be no significant changes in land use/land cover from 2015 to the end of the 21st century. By comparing the land use/land cover classification maps of the XRB, it was found that the spatial distribution of land use/land cover in the two periods were different, especially the type of urban area. The urban area increased significantly due to deforestation and the conversion of cultivated land. The urban area in 2015 and in the future will be increased by 547% relative to that at the baseline period. The forest and cultivated land areas in 2015 and the future will be decreased by 2.94% and 13.35% relative to that at the baseline period, respectively.

### 3.3. Results of Sensitivity Analysis and Model Performance Assessment

Table 2 lists the results of the global sensitivity analysis by using SWAT-CUP, based on the sensitivity ranking of the parameters. For the simulated streamflow, CN2, CH_K2, SOL_Z, SURLAG, ESCO, GW_DELAY, GWQMN, SOL_K, CANMX, SOL_AWC, ALPHA_BF, and CH_N2 were the first 12 high sensitivity parameters, while USLE_P, SLSUBBSN, BIOMIX, SPEXP, and SPCON were the top five high-sensitivity parameters for the simulated sediments. In the streamflow parameters, the SCS runoff curve number 'CN2' ranked first, much higher than the others. For a given catchment, CN2 controls the main runoff confluence process and represents the confluence capacity of different underlying surfaces. In the sediment parameters, the USLE (Universal Soil Loss Equation) equation support

practice factor 'USLE_P' is the most sensitive, which indicates the ratio of soil loss under soil and water conservation measures to soil loss under corresponding slope conditions. Table 2 shows that parameters representing the surface runoff, soil properties, groundwater return flow, ground water, and land cover management are sensitive. Consequently, it is important for the streamflow and sediment simulation to accurately estimate these parameters.

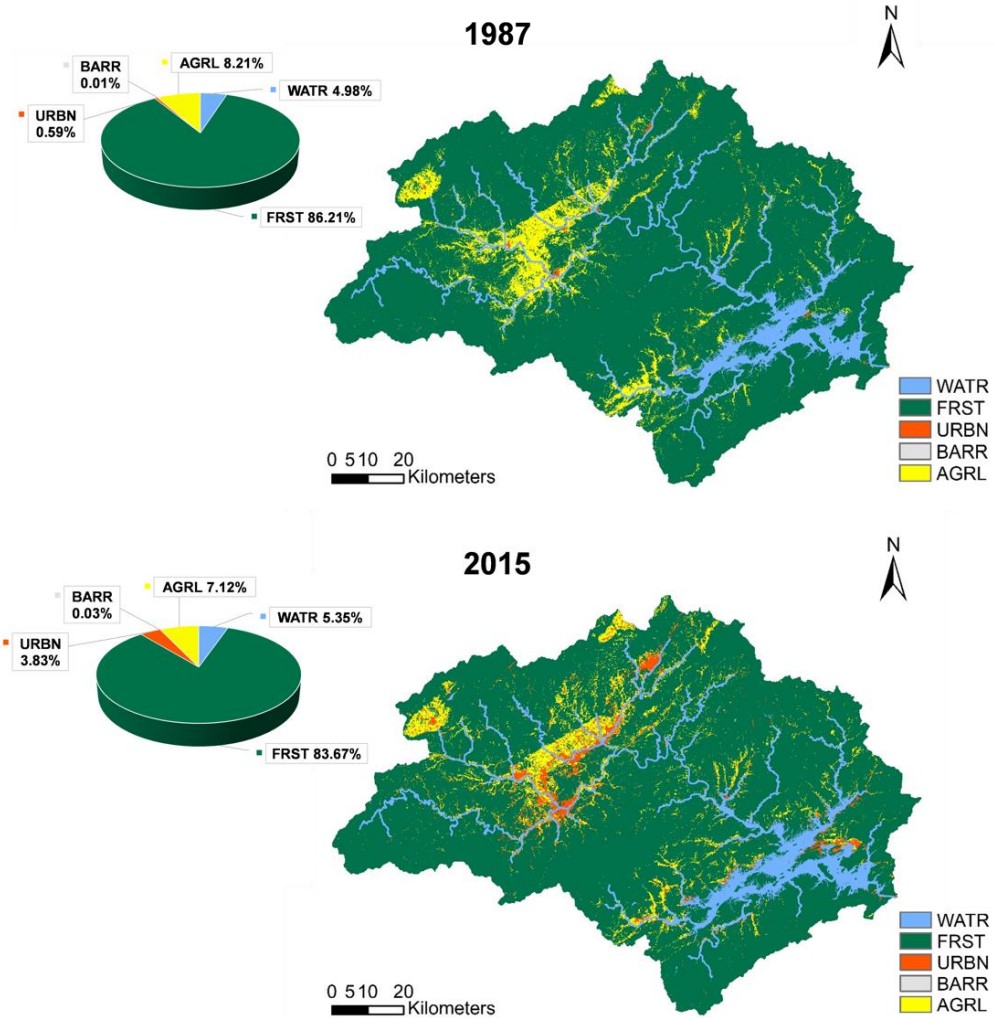

**Figure 4.** The land use/land cover classification maps in 1987 and 2015 under the $SN$ and $SN^{LC}$ scenarios and the proportion of area of each land use/land cover type. The land use/land cover in 1987 and 2015 represent the land use/land cover in the simulation baseline period (1973–2005) and the future (2068–2100), respectively. WATR indicates water body, FRST indicates forest, URBN indicates urban land, BARR indicates bare land, AGRL indicates cultivated land.

The SWAT model was calibrated and validated on a monthly scale in 2001–2010 and 2011–2014 for the streamflow for two stations (Tunxi and Yuliang station), respectively. The results are shown in Figure 5 and Table 3. For Tunxi Station, the observed and simulated streamflow fit well with the values of $NSE = 0.83$ and $r^2 = 0.85$ for the calibration period and $NSE = 0.89$ and $r^2 = 0.90$ for the validation period. For Yuliang Station, the observed and simulated streamflow were in satisfactory agreement with values of $NSE = 0.64$ and $r^2 = 0.69$ for the calibration period, and $NSE = 0.73$ and $r^2 = 0.88$ for the validation period. Based on the whole period of sediment monitoring data (2006–2014), the model was calibrated and validated on a monthly scale in 2006–2012 and 2013–2014 for the sediment for the same stations, respectively (see Figure 6 and Table 3). $NSE$ and $r^2$ between the observed and simulated sediment transport rate were larger than 0.60 and 0.62 for the calibration period for two stations. The results of the model validation at a monthly time

step were good ($NSE$ = 0.74 and $r^2$ = 0.81) for Tunxi Station and satisfactory ($NSE$ = 0.47 and $r^2$ = 0.55) for Yuliang Station. The hydrological model captured the low and some of the peak values of the flow and sediment very well. Overall, the SWAT model was mainly satisfied with the observed data in the XRB. That is to say, it is acceptable to use the calibrated parameters incorporated with the SWAT database for further simulations.

**Table 2.** The results of the sensitivity analysis and calibration for the SWAT model.

| Parameter | Definition | Sensitivity Analysis | | Calibration | | |
|---|---|---|---|---|---|---|
| | | *t*-Statistics | *p*-Value | Min | Max | Optimal |
| Streamflow | | | | | | |
| CN2 * | SCS runoff curve number for moisture condition II | −35.47 | 0.00 | −0.5 | 0.5 | 0.047 |
| CH_K2 | Effective hydraulic conductivity in main channel alluvium (mm/h) | −2.64 | 0.01 | −0.01 | 500 | 378.873 |
| SOL_Z * | Depth to bottom of first soil layer (mm) | 2.57 | 0.01 | −0.5 | 0.5 | 0.148 |
| SURLAG | Surface runoff lag time (days) | −0.96 | 0.34 | 0.05 | 24 | 17.324 |
| ESCO | Soil evaporation compensation factor | 0.64 | 0.53 | 0 | 1 | 0.347 |
| GW_DELAY | Groundwater delay (days) | 0.58 | 0.56 | 30 | 450 | 62.025 |
| GWQMN | Threshold depth of water in the shallow aquifer for return flow to occur (mm $H_2O$) | −0.50 | 0.62 | 0 | 5000 | 46.250 |
| SOL_K * | Saturated hydraulic conductivity of first soil layer (mm/h) | 0.39 | 0.70 | −0.8 | 0.8 | 0.638 |
| CANMX | Maximum canopy storage (mm $H_2O$) | 0.29 | 0.77 | 0 | 100 | 90.425 |
| SOL_AWC * | Available water capacity of first soil layer (mm/mm) | 0.19 | 0.85 | −0.5 | 0.5 | −0.251 |
| ALPHA_BF | Baseflow alpha factor (days) | −0.14 | 0.89 | 0 | 1 | 0.768 |
| CH_N2 | Manning's "n" value for the main channel | −0.01 | 0.99 | −0.01 | 0.3 | 0.295 |
| Sediment | | | | | | |
| USLE_P | USLE equation support practice factor | −39.88 | 0.00 | 0 | 1 | 0.020 |
| SLSUBBSN * | Average slope length (m) | −12.38 | 0.00 | −0.9 | 0.9 | −0.498 |
| BIOMIX | Biological mixing efficiency | −6.52 | 0.00 | 0 | 1 | 0.051 |
| SPEXP | Exponent parameter for calculating sediment re-entrained in channel sediment routing | 1.07 | 0.29 | 1 | 1.5 | 1.429 |
| SPCON | Linear parameter for calculating the maximum amount of sediment that can be re-entrained during channel sediment routing | 0.77 | 0.44 | 0.0001 | 0.01 | 0.006 |

Note: * The asterisk means the existing parameter value is multiplied by (1+ a given value).

This study used the SUFI-2 approach to analyze the sediment uncertainty, mainly resulting from the uncertainties in the CMIP5 GCM projections and land use/land cover information. In SUFI-2, the parameter uncertainty, described by a multivariate uniform distribution in a parameter hypercube, accounted for all sources of uncertainties in the hydrological model. The propagation of parameter uncertainty led to the output uncertainty, which was quantified by the 95% prediction uncertainty (95PPU) band. Latin hypercube sampling was used to calculate the 95PPU at the 2.5% and 97.5% levels of the cumulative distribution function of the output variables [48]. Two indices, the *p*-factor (the percent of observations bracketed by the 95PPU) and *r*-factor (the relative width of 95% probability band), were calculated to evaluate the goodness of calibration uncertainty on the basis of the *p*-factor approaching 100% and the *r*-factor approaching 1. For streamflow, it is considered to be satisfactory if the *p*-factor >70% while having an *r*-factor of around 1 [47,48]. For the sediment, a smaller *p*-factor and a larger *r*-factor could be acceptable (SWAT-CUP user-manual). In this study, the 95PPU of streamflow brackets was 88% of the observations for Tunxi Station and 76% of the observations for Yuliang Station, while the *r*-factor equaled 1.01 and 1.25, respectively. The uncertainty analysis results of the sediment showed that the 95PPU bracketed 51% and 39% of the observations for Tunxi and Yuliang Stations, respectively. Meanwhile, the *r*-factor equaled 0.62 for Tunxi and 0.51 for Yuliang, which are very close to a suggested value of 1.

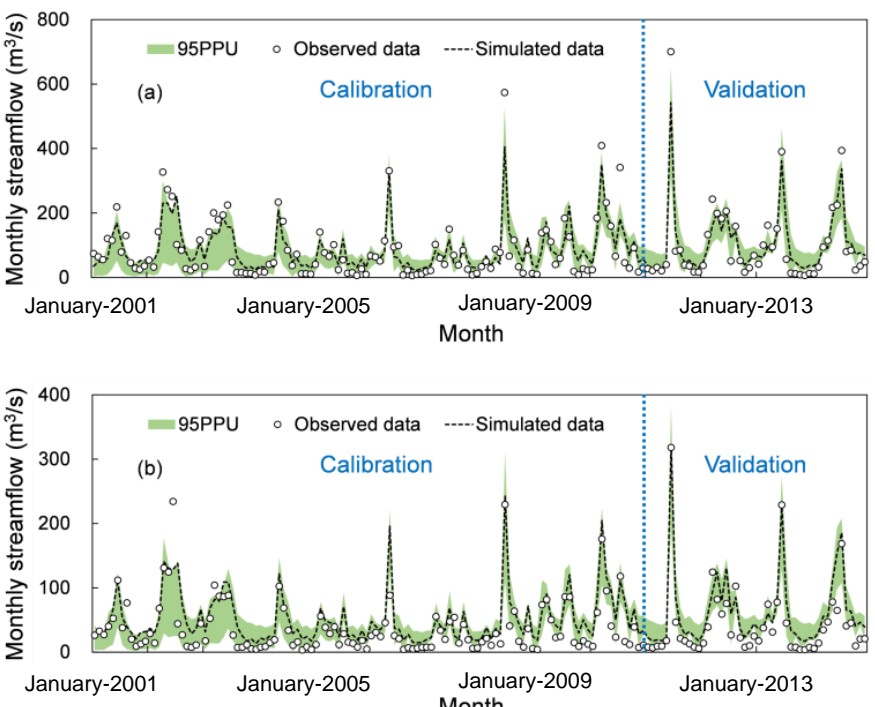

**Figure 5.** A comparison between the observed and modeled monthly streamflow in XRB during the calibration (2001–2010) and validation (2011–2014) periods. (**a**) The comparison result for Tunxi Station; (**b**) the comparison result for Yuliang Station.

**Table 3.** The SWAT performance during the calibration and validation periods.

| Variables | Stations | Periods | Monthly Average | | *NSE* | $r^2$ | Classes |
|---|---|---|---|---|---|---|---|
| | | | Observed | Simulated | | | |
| Streamflow (m³/s) | Tunxi | Calibration | 78.3 | 66.6 | 0.83 | 0.85 | Very good/Good |
| | | Validation | 108.9 | 114.7 | 0.89 | 0.90 | Very good/Good |
| | Yuliang | Calibration | 35.7 | 38.8 | 0.64 | 0.69 | Satisfactory/Satisfactory |
| | | Validation | 46.2 | 67.7 | 0.73 | 0.88 | Good/Good |
| Sediment (thousand tons) | Tunxi | Calibration | 38.9 | 43.3 | 0.70 | 0.71 | Good/Satisfactory |
| | | Validation | 38.1 | 54.0 | 0.74 | 0.81 | Good/Good |
| | Yuliang | Calibration | 16.0 | 24.1 | 0.60 | 0.62 | Satisfactory/Satisfactory |
| | | Validation | 28.0 | 29.8 | 0.47 | 0.55 | Satisfactory/Satisfactory |

### 3.4. Separating Impacts of Climate Variability and Land Use/Land Cover Change on Sediment

In the section of the estuary into the Xin'anjiang Reservoir, Jiekou, the mean annual transported sediment was $48.93 \times 10^4$ tons/yr at the baseline (1973–2005). In the future period (2068–2100), the mean annual transported sediment will be $69.53$–$76.31 \times 10^4$ tons/yr, with a variation of 42.10–55.97% relative to that in the baseline period (combining effects of climate change and land use/land cover change). We quantified the contribution of climate change and land use/land cover change impacting the transported sediment at the mean annual scale by using the framework described in Section 2.3.3. The results showed that the joint climate and land use/land cover changes caused an increase in the mean annual transported sediment of $20.60$–$27.39 \times 10^4$ tons/yr (see Table 4). The mean annual transported sediment is expected to increase under both the individual and combined climate and land use/land cover change impacts. Changes in the mean annual transported sediment will be mainly driven by climate change if the land use/land cover conditions in the future are kept as in 2015. In this case, the land use/land cover change might weaken the influence on sediment attributed to climate change. Specifically, the increases in the annual transported sediment for the predication period (2068–2100) due to climate variability

are 20.07–26.85 × 10⁴ tons/yr, which represent a contribution of 97.45–98.05%, while the land use/land cover change will lead to an increase in the annual transported sediment by 0.53–0.60 × 10⁴ tons/yr, with a contribution ranging from 1.95% to 2.67%. RCP8.5 showed smaller effects in increasing the influence on the sediment attributed to climate change than RCP2.6 and RCP4.5. However, the results may be very different if the future land use/land cover condition changes significantly compared with 2015.

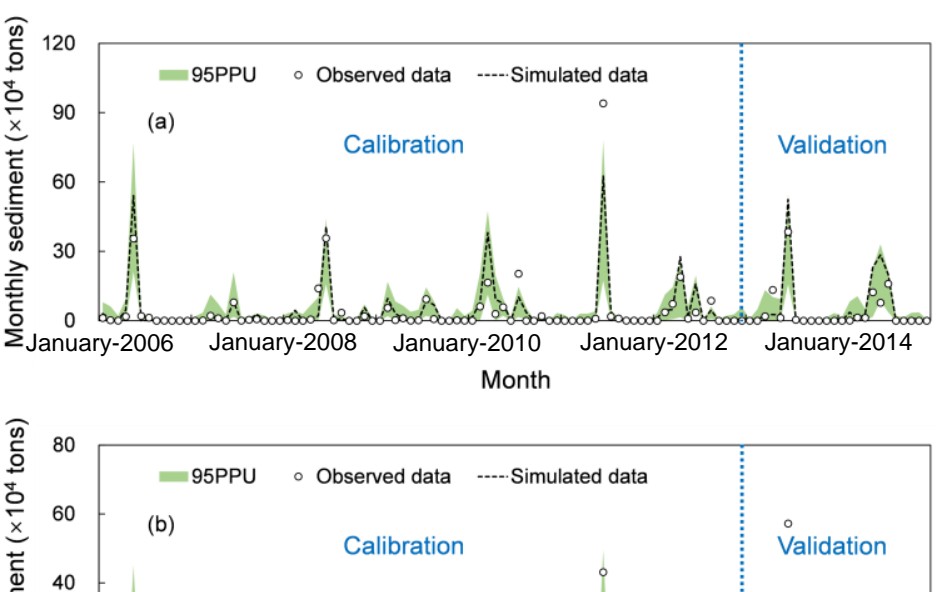

**Figure 6.** The comparison between the observed and modeled monthly transported sediment in XRB during the calibration (2006–2012) and validation (2013–2014) periods. (**a**) The comparison result for Tunxi Station; (**b**) the comparison result for Yuliang Station.

**Table 4.** The results of separating the impacts of climate change and land use/land cover change on the sediment in the XRB.

| Scenarios | $\Delta D_{RCPj}(10^4$ tons) | $\Delta D^{LCj}(10^4$ tons) | $\Delta D^{LC}_{RCPj}(10^4$ tons) | $\alpha_{RCPj}(\%)$ | $\alpha^{LCj}(\%)$ |
|---|---|---|---|---|---|
| RCP2.6 | 26.85 | 0.53 | 27.39 | 98.05 | 1.95 |
| RCP4.5 | 20.07 | 0.53 | 20.60 | 97.45 | 2.55 |
| RCP8.5 | 21.87 | 0.60 | 22.47 | 97.33 | 2.67 |

The spatial distribution of the sediment yield (sediment from the sub-basin that is transported into the reach during the time step) in the baseline period (with the land use/land cover map in 1987) and the relative variation ratio of the sediment yield under RCP2.6, RCP4.5, and RCP8.5 are shown in Figure 7. The relative variation ratio is described as a percentage of sediment yield in the baseline period. This means that we used the difference between the modeling value of the sediment yield in different RCPs and at the baseline as a numerator and the mean value of the sediment yield in the baseline period as a denominator. In the baseline period, the sediment yields from each sub-basin were 0.02–2.07 tons/ha/yr, with an average value for the whole XRB of 0.75 tons/ha/yr. In future scenarios (RCP2.6, RCP4.5 and RCP8.5), sediment yield had a strong response to climate change. Compared to the baseline period, our modeling analysis predicted dramatic increases in the sediment yield for each sub-basin under all three RCPs, especially under

RCP2.6 (with an increase of 19.20–85.70%) and RCP8.5 (with an increase of 34.18–68.05%). The increases in future sediment yield under the scenario of RCP2.6 were mainly concentrated downstream of the basin and in the area around Xin'anjiang Reservoir, while those under the scenario of RCP8.5 were mainly concentrated in the upstream of the basin. This is mainly because the precipitation significantly increases from April to July, which is the cultivation season of the main crops in the XRB, under the scenario of RCP8.5. The area of cultivated land in the upstream sub-basins of Jiekou is relatively larger than that in the downstream sub-basins. Consequently, frequent farming activities lead to an increase in the sediment loss with rainfall runoff, increasing the sediment input of the reservoir. It can be found that the spatial distribution of sediment yield change is consistent with that of water yield change, which is the net amount of water that leaves the sub-basin and contributes to streamflow in the reach.

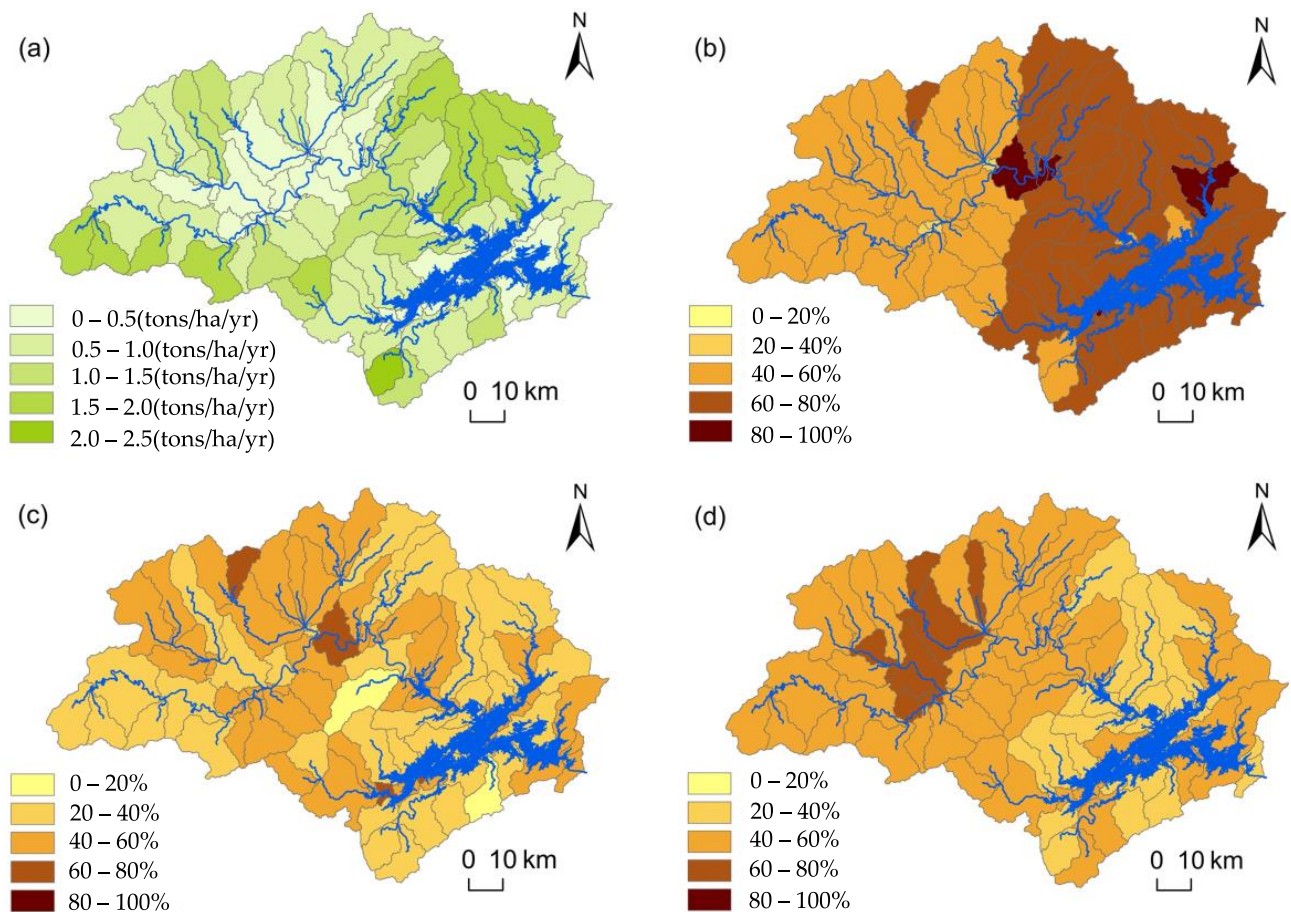

**Figure 7.** The spatial characteristics of annual sediment yield in XRB. (**a**) The mean annual sediment yield of every sub-basin in the baseline period. (**b**) The relative changes in the sediment yield between RCP2.6 and baseline. (**c**) The relative changes in the sediment yield between RCP4.5 and the baseline. (**d**) The relative changes in the sediment yield between RCP8.5 and the baseline.

Figure 8 shows the inter- and intra-annual variability in sediment transported with water out of reach in Jiekou under the baseline period, RCP2.6, RCP4.5, and RCP8.5. It can be seen that the inter-annual variation in the transported sediment in Jiekou is significant. Compared to the baseline, the mean annual transported sediment under RCP2.6, RCP4.5, and RCP8.5 increased dramatically by 40.91–54.75% when there was no land use/land cover change from 1987 to the future. The largest increase in the mean annual transported sediment is under scenario RCP2.6, followed by scenario RCP8.5. Through correlation analysis, the annual transported sediment was positively correlated with rainfall (*r* is 0.72 in the baseline period; *r* is 0.58–0.73 under different RCPs) and runoff (*r* is 0.77 in

baseline period; *r* is 0.65–0.91 under different RCPs), respectively, indicating that rainfall and runoff have a great impact on sediment output. Meanwhile, the nonparametric Mann–Kendall test [57,58], commonly used to assess the significance of trends in hydrology and climatology, was used to detect trends in the time series of the annual transported sediment. The annual transported sediment (2068–2100) exhibited a positive trend for the Jiekou site at the $\alpha = 0.1$ level of significance under RCP2.6 and RCP4.5, respectively, but no significant trend under RCP8.5 ($\alpha = 0.1$).

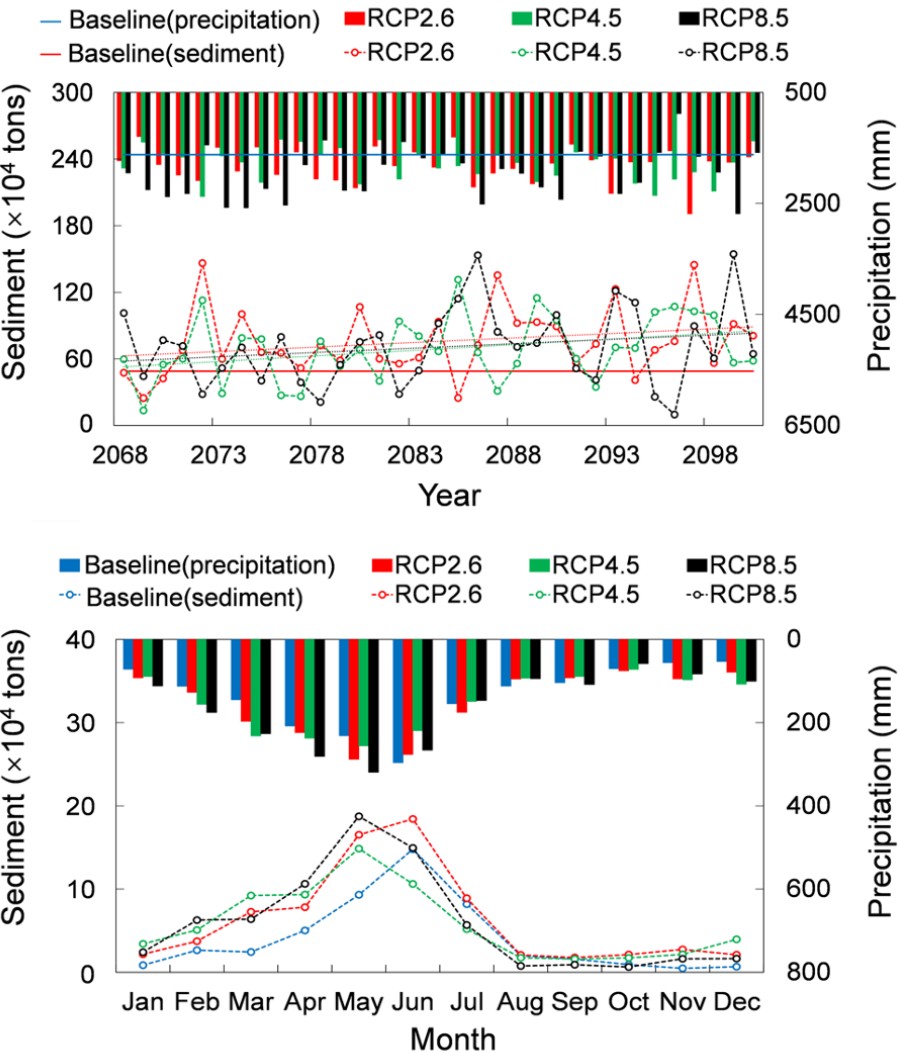

**Figure 8.** The inter- and intra-annual (monthly) variability in the sediment transported with water out of reach at Jiekou (estuary of Xin'anjiang River) under different climate scenarios.

A non-uniform distribution of the mean monthly transported sediment in Jiekou shows that the transported sediment in spring and summer accounted for 85.36% of the total sediment output in 1973–2005 and 73.95–80.88% in 2068–2100. Under scenario RCP2.6, the mean monthly transported sediment exhibited a significant increase in March–June, but no obvious change in June–February of the following year. Under scenario RCP4.5, the mean monthly transported sediment increased significantly in January–May but decreased in June–August. Under scenario RCP8.5, there was a significant increase in February–May, but a slight decrease in June–October. The intra-annual (monthly) distribution of the transported sediment is consistent with that of rainfall. It is indicated that the transported sediment is not only related to rainfall intensity, but also to the time distribution of rainfall.

*3.5. Implication for Water Quality Management of Reservoir/Lake*

Influenced by the temperate monsoon climate, more than 60% of the annual precipitation was recorded in April–August in XRB. According to the CMIP5 outputs, the monthly precipitation increased significantly in June and the frequency of heavy rainfall events increased in the flooding season under the three RCP scenarios. Correspondingly, the reservoir's inflow volume increased sharply after heavy rainfall. Compared with small and medium rainfall, it is easier for heavy rainfall or rainstorms to cause massive soil erosion. Seventeen HJ-1 A/B images during 10 heavy rainfall events from 2009 to 2014 were used to illustrate the relation of total suspended matter (TSM) concentration in the estuary of the Xin'anjiang River to the amount of precipitation of the basin. A significant positive correlation could be found between the TSM concentrations and rainfall amount ($p < 0.005$) [41].

In the Xin'anjiang Reservoir, significant turbid density flow always follows heavy rainfall events and rainstorms, which affects the reservoir water quality, especially in Jiekou estuary. It was investigated that the first small peak flow in March 2018 in Jiekou caused great changes in the water transparency and nutrient concentration, which indicates that the first peak inflow discharge of each year has a great impact on the water quality of the reservoir [33]. The particulate matter contributes to most nutrient inputs, which means that heavy rainfall events could lead to very high nutrient input into the reservoir/lake due to massive erosion from the upstream catchment and the area surrounding the reservoir/lake [59]. A large number of external nutrients carried by heavy rainfall (or rainstorms) and floods as well as the sediment resuspension caused by flood scouring increase the nutrient concentration of the water body in the reservoir/lake [60]. The degrees of eutrophication are aggravated correspondingly and suitable conditions for algae growth are provided. Therefore, understanding the effects of rainfall increases in the flooding season (especially the frequency of heavy rainfall events or rainstorm increases) on sediment yield in the basin could help water managers to strengthen the management of heavy rainfall runoff. It is also advantageous to the protection of water environment for reservoirs/lakes.

## 4. Conclusions

This study demonstrated that the sediment load and streamflow of XRB would significantly increase in the future under the integrated impacts of climate change and land use/land cover change. Sediment generated from the sub-basins above the Jiekou section (transported into the reservoir) will increase by 42.10–55.97% in 2068–2100, relative to that in the baseline period. Rainfall and temperature are the major climatic affecting factors in these increases, and the land use/land cover change can be attributed to the deforestation and urbanization during the simulating period. We found that more than 90% of these increases in sediment will be caused by climate change if the land use/land cover situation in the future are not obviously changed. While climate change combined with land use/land cover change in all three RCPs projected an increase in sediment, there were disagreements on the spatiotemporal distribution of sediment yield under multiple scenarios. In terms of space, the increases in the future sediment yield are mainly concentrated in the downstream of the basin under RCP2.6 but in the upstream of the basin under RCP8.5. In terms of time, more precipitation and floods in the wet season may occur in the future. Consequently, this will increase the sediment yield by 22.07–46.12% in the wet season (March–August) with respect to the baseline scenario. Therefore, it is important to emphasize increasing adaptation to climate change and land use/land cover change when designing and managing water environmental resources of the reservoirs and catchments.

In summary, climate change and land use/land cover can exert a great influence on the sediment yield in this humid and monsoonal climate region with separated or combined effects. The projections of future changes in sediment yield suggest the great challenge that lakes or reservoirs will face, because increasing sediment yield is associated with the high input of nutrients, especially phosphorus, which is a critical element for phytoplankton proliferation and algal bloom occurrence. Our findings can greatly benefit

managers/decision-makers in improving their understanding of these effects on rainfall–runoff processes and soil erosion as well as nutrient delivery in the catchment. Moreover, it can help them to design and adopt reasonable measures for watershed management and local governments regarding environmental conditions including climate change and land use/land cover change.

**Author Contributions:** Conceptualization, B.Q. and L.L.; Methodology, H.L.; Software, H.L.; Validation, H.L., Y.L., and J.J.; Formal analysis, H.L. and Y.L.; Investigation, C.Y. and Z.W.; Resources, Y.L. and C.Y.; Data curation, J.J. and C.Y.; Writing—original draft preparation, H.L.; Writing—review and editing, B.Q. and L.L.; Visualization, K.S. and G.Z.; Supervision, B.Q. All authors have read and agreed to the published version of the manuscript.

**Funding:** This research was funded by the National Natural Science Foundation of China (42171034, 41671205 and 41830757) and Yunnan University (C176220100043).

**Institutional Review Board Statement:** Not applicable.

**Informed Consent Statement:** Not applicable.

**Data Availability Statement:** The relevant data was available on the request of corresponding authors.

**Conflicts of Interest:** The authors declare no conflict of interest.

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
