# Peer review of "Modeling the Effects of Climate Change and Land Use/Land Cover Change on Sediment Yield in a Large Reservoir Basin in the East Asian Monsoonal Region"

_water, doi:10.3390/w14152346_

Round 1

Reviewer 1 Report

The manuscript titled as " Modelling the effects of climate change and land use/cover change on sediment yield in a large reservoir basin in the East Asian monsoonal region" by Li et al.

The authors quantified the sediment yield of the Xin’anjiang Reservoir Basin in response to future climate change and land use/cover change under different scenarios. The methodology and results were well presented in the manuscript. I would recommend that the following minor criticisms be considered.

1) The objective(s) should be clearly stated at the end of Introduction section.

2)  Table 3 is missing in the manuscript.

Reviewer 2 Report

Dear Authors,

Please see the section-wise comments below on your research article with the following details.

Manuscript title: Modelling the effects of climate change and land use/cover change on sediment yield in a large reservoir basin in the East Asian monsoonal region

Manuscript Number: water-1771017

Journal Submitted: Water

Specific Comments:

Title:

The title is representative of the study.

Abstract:

The abstract is very well-written and represents the study results in a way that can be understood. However, the first half is unnecessarily dedicated to the foundation and introduction of the study.

I do not find the results are something novel.

L 35-37: Why after 2015? Why not from the current year?

The conclusions are missing altogether and what is the implication of this study?

Keywords are less in number.

Please modify if it can be applicable at some regional to global level settings.

I do not see the results and study a novel case study.

Introduction:

L 53-54: Why would you discuss the streamflow and sediment yield more than reservoir sediment yield? Please explain it more.

L 54-56: Improper citations. This is rampant in the whole of MS.

The literature review is not extensive and reliable. It needs to be enriched with local and exotic case studies as well.

Further, the authors need to explain their objectives. The objectives are missing altogether.

Materials and Methods:

Figure 1 must be provided with higher resolution.

The study design is appropriate to answer the questions.

Methods are the strongest part of this manuscript.

Results:

Figure 2 caption shows that the authors are highly nonserious.

Figure 2 should go to the supplementary section.

The results are described in little detail while there could be more details on this topic.
This can make the data presented in a way that is readable on a wider scale.

Discussion:

It is hard for me to distinguish the discussions from the results. I feel that the discussion part is an extension of the results, and the discussions are not included. In this case, the best thing you can do is to merge the results and discussions into one section. By doing so, please ensure that the results are discussed from multiple angles and placed in context without being overinterpreted. This is not detectable in this section.
Conclusions

The conclusions are too long and dragged. I am sure the authors need to summarize this section and shift the main outcomes in the abstract and discussions section, wherever it can be suitable. One thing that needs to be incorporated is the future implication of this study and how it can be beneficial at the regional to global scale.

References:

There are several errors in in-text citations and reference format.

Concluding remarks:

This study provides a piece of baseline information on this topic which could be a good addition to the existing knowledge on this topic at the local level. The article is not consistent in itself, and the authors have provided simple and to-the-point explanations. However, I strongly feel that it is more relevant on the local scale. Furthermore, this study is a replication of recent studies on the same topic. The authors need to work harder and improve this manuscript by undertaking Major Revisions. One of the most important issues in the English language problem which shows that the authors have not written it in a way that could appeal to the broader readerships of Water. 

Reviewer 3 Report

The language of this paper requires review by an expert English speaker. The meaning of sentences is often lost by unclear use of language. There is also too much information and a listing of data for the reader to make sense of or follow a logical flow of the paper. It is not necessary or useful to include every step, equation, or piece of data in the Methods and Results. The most important things to clearly communicate are why the research is important, the basic procedure for performing the study, and the most useful pieces of information gained - in language that can be easily understood by someone, not in your specific field.

Round 2

Reviewer 3 Report

The English language is improved as are the introduction and relevance.